# Modelling Protein Plasticity: The Example of Frataxin and Its Variants

**DOI:** 10.3390/molecules27061955

**Published:** 2022-03-17

**Authors:** Simone Botticelli, Giovanni La Penna, Germano Nobili, Giancarlo Rossi, Francesco Stellato, Silvia Morante

**Affiliations:** 1Dipartimento di Fisica, Università di Roma Tor Vergata and Sezione di Roma Tor Vergata, INFN, Via della Ricerca Scientifica 1, I-00133 Roma, Italy; simone.botticelli@roma2.infn.it (S.B.); germano.nobili@roma2.infn.it (G.N.); giancarlo.rossi@roma2.infn.it (G.R.); francesco.stellatof@roma2.infn.it (F.S.); silvia.morante@roma2.infn.it (S.M.); 2Istituto di Chimica dei Composti Organometallici, Consiglio Nazionale delle Ricerche, Via Madonna del Piano 10, I-50019 Firenze, Italy; 3Centro Fermi—Museo Storico della Fisica e Centro Studi e Ricerche E. Fermi, I-00184 Roma, Italy

**Keywords:** molecular statistics, high-performance computing, frataxin, iron homeostasis, cancer

## Abstract

Frataxin (FXN) is a protein involved in storage and delivery of iron in the mitochondria. Single-point mutations in the *FXN* gene lead to reduced production of functional frataxin, with the consequent dyshomeostasis of iron. FXN variants are at the basis of neurological impairment (the Friedreich’s ataxia) and several types of cancer. By using altruistic metadynamics in conjunction with the maximal constrained entropy principle, we estimate the change of free energy in the protein unfolding of frataxin and of some of its pathological mutants. The sampled configurations highlight differences between the wild-type and mutated sequences in the stability of the folded state. In partial agreement with thermodynamic experiments, where most of the analyzed variants are characterized by lower thermal stability compared to wild type, the D104G variant is found with a stability comparable to the wild-type sequence and a lower water-accessible surface area. These observations, obtained with the new approach we propose in our work, point to a functional switch, affected by single-point mutations, of frataxin from iron storage to iron release. The method is suitable to investigate wide structural changes in proteins in general, after a proper tuning of the chosen collective variable used to perform the transition.

## 1. Introduction

Friedreich’s ataxia (FRDA) is an autosomal-recessive genetic condition that causes ataxia, sensory loss and cardiomyopathy worsening over time [1,2,3]. It was first described by the German physician Nikolaus Friedreich in 1860. The cause of the disease is to be found in mutations of the *FXN* gene located on chromosome 9 which result in an abnormal expansion of a non-coding GAA triplet repeat in the first intron of the *FXN* gene, leading to a reduced transcription of the gene with consequent low levels of frataxin production [4]. Depending on the specific kind of mutation, a patient may end up with an insufficient level of frataxin, a nonfunctional frataxin or frataxin that is not correctly localized in the mitochondria [5,6].

The precise physiological role of frataxin is not totally clear [7], although it is known to be associated with iron trafficking and homeostasis. In particular, frataxin assists iron–sulfur protein synthesis in adenosine triphosphate generation processes and is involved in the regulation of iron transfer in the mitochondria [8]. Frataxin deficiency leads to iron overload in the mitochondria with the creation of unsafe excess of reactive oxygen species [2].

Frataxin synthesis takes place in cytoplasmic ribosomes starting with a large precursor molecule with mitochondrial targeting sequences. Upon entering the mitochondria, the molecule undergoes a proteolytic reaction, yielding mature frataxin [9]. Mature frataxin is a ∼170 long amino acid protein, with the first 40 residues used mostly for the transport across the mitochondrial membrane.

X-ray crystallographic data of isolated human frataxin (FXN, hereafter) 88-210 isoform (PDB 1EKG [10]) display the structure of residues 90-208. FXN has a rather compact structure consisting of a planar β-sheet formed by four β-strands supporting two almost parallel α-helices. FXN homologues in other species are structurally similar, except for the length of turns and the terminal tails extending from the ends of helices [11]. Human FXN has a tail sequence longer than what is found in bacteria and yeast. The N-terminus residues 81-89 of frataxin (FXN, hereafter) are structurally disordered. The length of the disordered N- and C-terminal chains have a strong impact on FXN thermal stability [12,13].

More recent information concerning FXN is the role of variants in cancer. Indeed, upon inspection of the catalogue of somatic mutations in cancer (COSMIC database [14]), missense variants are found in multiple human cancer tissues [15]. This is expected, because of the involvement of FXN and mitochondria in the control of oxidative metabolism [16].

Given the key role of FXN point mutations on its production level, ability to interact with iron and to correct deliver Fe to iron–sulfur (FeS) clusters, it is of the utmost importance to understand and characterize the impact of the various point mutations on the folding of the protein. In this respect, FXN represents a paradigmatic example of protein structural plasticity as a function of the sequence.

The present work is a further step after the previous bioinformatic [13,17] and simulation works [12,15,18] to identify the amino acids relevant to iron and protein binding, respectively.

In particular, in this paper we study the stability properties of the folded protein against a selected set of eight point mutations involved in cancer. We compute folding free energy differences and compare the results of the wild-type (WT) protein to those of the point-mutated ones. The theoretical analysis relies on the use of a very effective computational method, devised to enhance phase-space sampling for atomistic macromolecular models. The method is based on extending the scope of NpT Molecular Dynamics (MD) with the features of the so-called altruistic metadynamics.

A comparison of the computed free energy differences with data coming from thermodynamic stability measurements [15] shows that the method can be used to distinguish between single-point mutations that either stabilize or destabilize the native sequence. The results of the proposed method are also compared to recent bioinformatic predictions and to the few atomistic models reported in the literature.

## 2. Results

Before reporting the results of the simulations, we define the structural parameters sampled by our model with reference to the available experimental information.

In Figure 1 residues 90-210 in the folded structure of wild-type FXN (PDB 1EKG) are highlighted. FXN is characterized by a small β-sheet composed by five short anti-parallel β-strands (represented as yellow ribbons). This scaffold hosts two α helices (purple ribbons), located in the N- and C-terminal regions. The first α-helix is in region 92-114, α1 hereafter, and contains 3 Asp, 6 Glu, and only 1 Arg residues, thus indicating a high negative charge, at experimentally studied conditions (pH = 7.4), concentrated within a relatively short and rigid protein segment. The stability of the protein increases by salt addition [11], thus showing that electrostatic interactions play an important role in structural stability. In the figure, we have emphasized negatively charged carboxylate groups and positively charged groups as, respectively, red and blue spheres. Centers of red spheres are at Cγ(Glu), Cδ(Asp), and C terminus. Centers of blue spheres are at terminal N, Nζ(Lys), and Nη(Arg). It can be noticed the crowding of negative charges over and around the α1 motif. The eight mutated residues found in the COSMIC database and investigated in this work are emphasized in green. The single-point mutations are: D104G, A107V, F109L, Y123S, S161I, W173C, S181F, S202F (we use in the text an abbreviated nomenclature of variants compared to the nomenclature recommended by the Human Genome Variation Society (HGVS, see Appendix A)). With the exception of S161I, the residues are, in the native folded structure, close to α1. The second α-helix (α2, region 182-193) is also involved in interactions with α1, mainly via a hydrophobic patch.

As explained in Section 5.7, two experimental structures are available for human FXN, and both were used in the preparation of the simulated walkers. Even though the structures are characterized by different resolutions, they contain information about two different contexts: The first resembles the FXN assembly with identical partners; the second resembles the FXN assembly into the complex where FXN delivers iron to other partners. The choice of these two structures as representative of two different contexts of FXN is discussed in more detail in Section 3. The aim of our model is to introduce the information of at least these two different contexts in one single multiple-walkers simulation of the protein monomer. The two structures are characterized by different structural parameters, even when the latter are simplified in terms of “coordination numbers” like the one in Equation (Equation 16). The parameter $S_{\beta ,1–4}$ is used as the collective variable CV, while $S_{\alpha ,1}$ is the result of the same Equation (Equation 16) applied to atomic pairs involved in the hydrogen-bonds characteristic of α1 helix (see Table 1). The structural parameters $S_{\beta ,1–4}$, $S_{\alpha ,1}$, $S_{\alpha ,2}$ are, respectively, 15.7, 16.1, 6.8 for 1EKG, and 10.0, 11.7, 6.6 for 5KZ5 (chain A). Therefore, the length of β-sheet and α-helix, is significantly smaller in the mitochondrial complex than in the truncated isolated form of FXN. As explained in Methods, we choose $S_{\beta ,1–4}$ as the handle (CV), to induce and monitor the protein unfolding.

The distribution of the chosen collective variable as it is obtained by the meta-statistcs used as reference in the maximal constrained entropy method, is displayed in Figure 2. The distribution is, in the long-time limit of the simulation and for an infinite number of walkers, flat. In practice, this is not the case because of the many statistical limitations of atomistic simulations: (i) a small number of walkers; (ii) construction of external bias in the 20-ns time range; (iii) usage of the external bias in the 10-ns time range. However, with the exception of S181F all distributions are similar, with a shallow well in the region of *s* around 8–9. We notice that values of CV ($S_{\beta ,1–4}$) in the range of that adopted by the mitochondrial complex FXN form (see above), are sampled.

The extent of unfolding will be analyzed in more detail below, by comparing each configuration with the reference folded structures available in the literature. The meta-statistics, that we recall is obtained with no bias reweighting, Equation (Equation 2), looks a reasonable reference meta-statistics for all sequences, because it is only slightly dependent on the sequence itself. In all cases, the interactions responsible for either the folding and unfolding of the protein variants are sampled with similar weight. As a representative unfolded configuration, in Figure 3 we display the secondary structure (colored ribbon) and the position of the mutated residues (in green) for a configuration with $S_{\alpha ,1}$ = 13, $S_{\alpha ,2}$ = 6.6, and $S_{\beta ,1–4}$ = 2. It can be noticed that the two α-helices are only slightly changed in length, but, because of the unfolding of the β-sheet scaffold, the two helices change the relative orientation.

**Figure 3 molecules-27-01955-f003:**
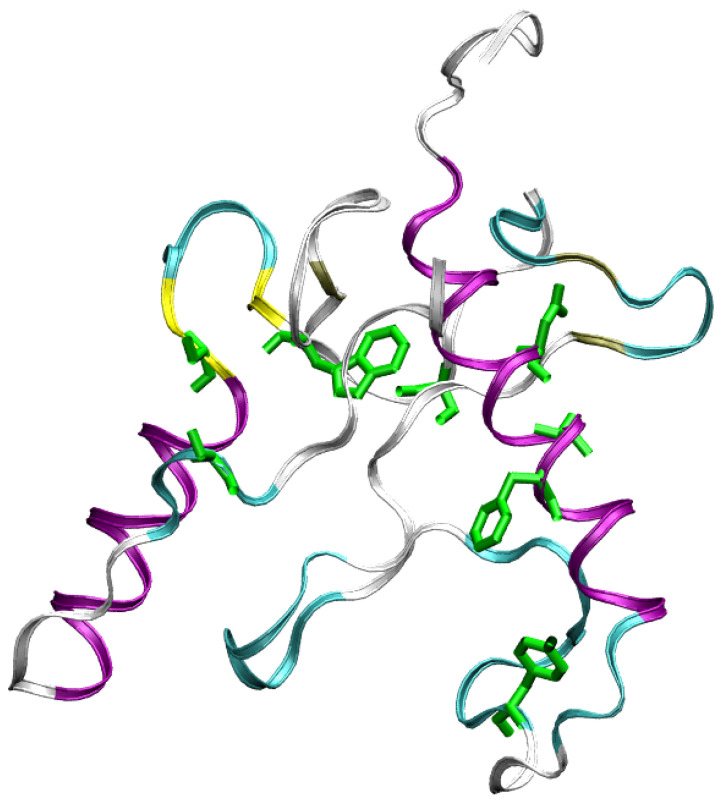
Structure of FXN, WT sequence, with $S_{\alpha ,1}$ = 13, $S_{\alpha ,2}$ = 6.6, $S_{\beta ,1–4}$ = 2 (see also Table 2). The drawing is made as in Figure 1.

The free energy as a function of the average of the chosen collective variable is displayed in Appendix A. Each panel is for a given protein sequence. Top panel is for the native wild-type (WT) sequence. In all functions, the zero of free energy is set to $S_{\beta ,1–4}=s$ = 14. We assume that in all cases the native-like folded state is the most stable and the value of *s* = 14 is sampled in meta-statistics for all sequences with similar weight (Figure 2). The stability of the native state has been shown by the experimental characterization (CD spectra) along with temperature, with the exception of W173C [15].

The curves do not depend on the parameter *T* in Equation (Equation 10) up to *T* = 400 K. This means that the contribution of ${\bar{S}}_{c}$ to the free energy is small. The force-field parameters used in all simulations do not hold so far away from room temperature, and it is not reasonable to extend *T* beyond 400 K. Only in the cases of S161I and S181F the complete unfolding is spontaneous, with a free energy change of zero when *s* achieves values of 2. The native sequence is that showing the largest stability. Variants F109L, S202F, W173C, and Y123S show almost the same behaviour of a lower stability compared to the native sequence. D104G is the single variant that shows a slightly larger resistance to unfolding, after a more stable state is achieved with *s* = 13.

The same free energy curve displayed in Appendix A is displayed in Appendix A by using the mean-field approximation for protein-solvent interactions (see Section 5). The free energy change in the two figures is very different in magnitude. The mean-field approximation is dominated by the energy contribution in all the range spanned by the collective variable *s*. However, we notice that the behavior of changes among sequences is the same of the more accurate free energy based on the constrained average of enthalpy *H* (Appendix A). The native sequence (WT) displays the largest positive change, thus reflecting the largest thermal stability of this sequence. All the variants display a smaller free energy change upon unfolding, with the noticeable exception of D104G, that is very similar to WT (see also Table 3 and Table 4, discussed below). Despite the thermal stability of all variants is overestimated by the mean-field approximation of protein-solvent interactions, the indication of a larger stability of WT and D104G compared to all other variants, observed in experiments, is confirmed.

The large difference in magnitude of the free energy change when computed with the mean-field approximation underlines the significant effect of detailed interactions between protein atoms and explicit water molecules and ions in the protein environment. The significance of these environmental effects is particularly relevant for some variants like S161I and S181F that, with the explicit solvent model, are almost flat, while in the mean-field approximation are steep. A detailed analysis of these effects will be discussed in a forthcoming article while, in the following, we assume, as for energetic analysis, the separation of energy contributions as in Equation (Equation 12).

On the basis of the analysis of metastatistics (Figure 2), we chose the value of s=〈CV〉 = 4 as the value corresponding to the unfolded state. This value is adequately sampled by all variants to a similar extent. Within the assumption that *s* = 14 and *s* = 4 are representative of folded and unfolded states, respectively, the summary of the results for unfolding free energy changes of the 8 studied single-point mutations of FXN with respect to the WT sequence is reported in Table 3.

The results of our proposed method are compared with several analysis reported in the literature [15,18]. As observed above, the maximal constrained entropy method (last two columns) is able to reproduce the fact that the D104G variant is the one showing a positive (column 6) or the least negative value (column 7). Remarkably, the positive value of ΔΔF obtained from Equation (Equation 12), is consistent with the increase of melting temperature of D104G with respect to WT, indicating the higher stability of D104G compared to all other variants. The detailed correct ranking of the decrease in stability of all variants except D104G is not exactly reproduced, but this issue is common to all methods used in the literature. The comparison between the method here proposed and similar methods reported in the literature is discussed in detail in Section 3, after the analysis of other structural parameters obtained by the simulations is described below.

The error estimated by the block analysis (see Section 5) is useful to assess the reliability of the result described above and analyzed in more detail below. In Table 5 we report the error affecting the quantity ΔF as it is obtained from Equation (Equation 12).

In the same table wer report the error affecting an important variable that is analyzed in the following, the solvent accessible surface area (SASA) and the error on the parameter λ as it is calculated in the fitting procedure (see Section 5). These data are reported for the wild-type sequence and for all states associated to a value of s=〈CV〉 in the range 2–14. The table shows that the error on ΔF is about 1%. The error on average SASA is smaller than the change of SASA as a function of the change of state *s*. The error on λ is very large, especially in those states with values of *s* intermediate between the mostly sampled configurations (see Figure 2). This behavior of λ is expected because the error on λ is calculated from $\frac{\partial s}{\partial \lambda }$ [21] and this quantity is large when λ = 0, that is when *s* is in the middle of the sampled values of CV. In this region the metastatistics provides the same average of CV as a sharp peak with the center at the average value. Therefore, this error on λ has a physical meaning related to the transition in CV that is imposed by the external bias. On the other hand, this behavior indicates that longer simulations are required to correctly sample the configurations that are intermediate between those mostly contributing to initial (folded) and final (unfolded) states. In the following we do not report the error for the studied quantities, assuming that the error is within 1%.

### 2.1. Structural Deviation with Respect to Reference FXN Structures

In this subsection we analyze the deviation of the protein structures collected with the meta-statistics with respect to known crystal structures of the native FXN sequence. We again emphasize that we use as reference: (i) the crystal structure of mature FXN form (segment 88-210, PDB 1EKG [22]) as it is obtained by X-ray diffraction; (ii) the crystal structure of a longer FXN segment (42-210), observed in the active mitochondrial complex (PDB 5KZ5 [23]) as it is obtained by electron microscopy (EM). We remind readers that the two structures, in the segment 90-210, are used as initial configurations, respectively, in walkers 1-60 (1EKG) and 61-90 (5KZ5) in the simulation here analyzed. The sequence 90-210 is used in the thermodynamic and biophysical studies [15] we compared to. The backbone root-mean square deviation (RMSD) (see Section 5) is a rough indication of the change in backbone structure RMSD values calculated between the 15 different models obtained for FXN in the 90-210 truncated form by solution NMR (PDB 1LY7 [24]) are in the range 1.0 ÷ 2.1 Å. The 12 FXN structures reported in the EM data (PDB 5KZ5 [23]) for FXN in region 90-210 display RMSD in the range 2.8 ÷ 3.3 Å. Therefore, a value of RMSD within approximately 3 Å indicates a structure close to the reference one within the lowest experimental resolution of EM. We indicate the deviations with respect to 1EKG and 5KZ5 as RMSD1 and RMSD2, respectively.

In Figure 4 we display the distribution of RMSD1 as it is obtained in the meta-statistics for all FXN variants. The sampling of RMSD1 values up to 18 Å$\tilde{c}$onfirms that unfolded configurations have been achieved with the method. Only in four cases (WT, A107V, D104G, and S161I) there is a visible sharp peak in the region with minimal RMSD. The area of this first peak is the probability of configurations almost identical to 1EKG. The values of probability are: WT 0.28 (RMSD1 < 3.3 Å); D104G 0.30 (RMSD1 < 3.1 Å); S161I 0.17 (RMSD1 < 2.4 Å); A107V 0.08 (RMSD1 < 2.3 Å). We notice that all of these configurations are provided by walkers that started from 1EKG configuration. This ranking shows that D104G variant enhances the chance to keep the structure of the native sequence (WT), mostly represented by configurations similar to 1EKG. All of the other variants display a broad distribution of RMSD1, showing that these variants are not constrained to resemble the WT structure by any energy barrier.

In Figure 5 we display the distribution of RMSD2 as it is obtained in the same meta-statistics for all FXN variants. In meta-statitiscs there is no visible amount of configurations resembling 5KZ5 structure (i.e., RMSD2 < 3 Å). This is consistent with the truncated sequence used in simulation (and thermal unfolding experiments). The 90-210 sequence lacks of those interactions responsible for the stability of FXN core into the active mitochondrial complex. Indeed, the RMSD of 5KZ5 with respect to 1EKG is 4.7 Å for the 90-210 backbone non-hydrogen atoms. The first and most intense peak in the distribution of RMSD2 is at about 5 Å for WT and all variants. Therefore, this peak in RMSD2 distribution contains the configurations resembling 1EKG, but with a significant contribution of those configurations significantly different from 1EKG. For WT, the contribution of the configurations with RMSD1 ≥ 3.3 Å is 11% of the configurations with RMSD2 < 5 Å.

To visualize the degree of structural disorder contained in the metastatistics, we display in Figure 6 254 structures uniformly sampled within those contributing to the first peak of RMSD1 distribution for WT. Only non-hydrogen backbone atoms are displayed for clarity. It can be noticed the high order in the α helices (see also Figure 1 that displays 1EKG structure approximately in the same orientation). The β-sheet is clearly softer, but the main scaffold of the protein is preserved by all of these configurations. Most of the structural disorder is concentrated in the C-terminal tail. This view is very similar to that obtained by comparing the different models obtained by NMR constraints in solution. However, a precise comparison with NMR distance restraints is out of the scope of this work.

Interestingly, the analysis of RMSD1 and RMSD2 distributions shows that the D104G variant has the most intense peak in the lowest RMSD region. This indicates that D104G enhances the chance to keep the 1EKG structure, together with a high propensity to keep minimal the structural distance with respect to 5KZ5 (RMSD2). This observation is consistent with the large positive ΔF for structure unfolding displayed in Appendix A: the D104G sequence is the variant to WT sequence most resistant to the breaking of the network of hydrogen bonds that keeps the integrity of the 1–4 β-sheet scaffold characterizing the resting FXN structure.

### 2.2. Changes in Molecular Surface upon Unfolding

The distribution of the solvent-accessible surface area (SASA) in the meta-statistics gives a first glance over the effects of the breaking and forming of critical hydrogen-bonds on the molecular stability. In Figure 7 the distribution of SASA in the meta-statistics is displayed for WT and all variants. Comparing this distribution with that of RMSD1 (Figure 4) and RMSD2 (Figure 5), we notice that the peak of *P* associated to the smallest SASA, in the range of 7000–8000 Å${}^{2}$, contains all structures closest to the 1EKG native structure and those different from 1EKG and closest to 5KZ5. The D104G and S202F variants display the largest populations for the most compact configurations (∼7000 Å${}^{2}$ of SASA). Noticeably, these two variants are more compact than the native sequence. Since this feature of SASA distribution is shared with the distribution of RMSD2, the structures closest to the mitochondrial assembly structure (5KZ5) are characterized by a small SASA, slightly smaller than WT. SASA of the 90-210 protein core in 1EKG and 5KZ5 is, respectively, 6726 and 8909 Å${}^{2}$. This means that the N-terminal chain of 5KZ5, that is removed in the simulations here reported, assists the expansion of the 90-210 protein core. Once the N-terminal chain is removed, the protein core becomes more compact. This compaction occurs more likely for D104G and S202F variants. However, the free energy change is for S202F less positive than for D104G and WT. Therefore, the mean-field force keeping configurations more compact is lower for S202F than for both WT and D104G.

### 2.3. Changes in Energy Components

In Table 4 we report the difference in the energy changes upon unfolding, measured by separating the energy components in our atomistic model and within the assumption of the averaging, at *T* = 300 K, of the protein-solvent contributions as in Equation (Equation 12).

In the second column, we report the changes in unfolding temperature measured by following the CD spectrum [15]. The values of ΔΔF, as shown in Appendix A, are all negative, with D104G approaching zero and in certain regions being positive.

The energy components more sensitive to the unfolding transition are the electrostatic and solute-solvent ones. These two components tend to balance each other in the change in potential energy with the protein unfolding. The last column contains the change of the sum of these two terms. The D104G variant displays a larger cancellation between the two terms, showing that the energy change is dominated by the van der Waals interactions. Because of the removal of significant electrostatic contributions in D104G, the increase in hydrophobic collapse dominates the D104G variant. In all the other cases the sum of electrostatic contributions (last column) is negative, thus assisting the unfolding by contrasting the sometimes positive van der Waals contribution.

Asp 104 displays in the crystal structure a large solvent accessibility assisted by the electrostatic repulsion between the several negatively charged groups concentrated in the α1 motif. Indeed, in the folded configurations, represented by the sampling corresponding to 〈CV〉 = 14, the difference of electrostatic energy between D104G and WT is −129.6 kcal/mol. The difference is positive for all the other variants, with the exception of S202F, that is −23.5 kcal/mol. The negative value of D104G is due to the removal of a negative charge in the α1 motif. More details about change in salt-bridge network is reported in the next sub-section. The difference in solute-solvent energy is, conversely, 206.5 kcal/mol for D104G with respect to WT, while the next larger value is 34.0 for S202F. The removal of the Asp 104 side-chain in D104G decreases significantly the hydration energy in the folded state, but the effect is largely compensated by the decrease in electrostatic repulsion. As we shall see below, the D104G variant is particularly effective in removing the electrostatic repulsion minimizing the perturbation of the α1 structural motif.

### 2.4. Changes in Secondary and Tertiary Structure

In our model, the parameters $S_{\alpha ,1}$ and $S_{\alpha ,2}$ describe the behavior of helices α1 and α2 upon the unfolding of the β-sheet (CV, that is $S_{\beta ,1–4}$). In Table 2, we report the average of these parameters after imposing to the meta-statistics the average of CV at, respectively, 4 and 14. We recall that these two average values for CV are representative of, respectively, unfolded and folded states. We first notice that the length of α2 helix is not perturbed by the length of β-sheet. On the other hand, α1 is often destabilized, with a decrease of 2 in the number of α-helix hydrogen bonds. The decrease is not very sensitive to the protein variant, but in our approach the work to make this change in protein structure is monitored by the free energy approximations summarized in Table 3 and displayed in Appendix A. This work is in all cases positive, with the larger values for WT and D104G variant. In some variants, S161I and S181F, and without approximating the protein-solvent interactions, the work is null.

A further structural parameter, $S_{SB}$, is used to monitor the state of charged groups in the protein (see Section 5 for the definition). The latter is more sensitive to changes in the tertiary protein structure. We notice that $S_{SB}$ is slightly larger when the protein is unfolded. This is the effect of the α1 helix, that, in the folded configurations, project the charged side-chains towards the solvent, preventing the formation of a network of electrostatic interactions. Conversely, when the β-sheet is demolished, many side-chains are allowed to neutralize the charges over the rigid α domains, decreasing hydration of side-chains in α1 and α2. This event is counterbalanced by the eventual dissociation of hydrophobic patches, the latter sealing the interface between the two α-helices. This patch is demolished when one helix rotates with respect to the other (see the analysis reported below). This effect is summarized by the changes in energy components reported in Table 4 and described above.

It must be noticed that in the D104G variant the number of salt-bridges is expected to be lower than in all other variants, because of the absence of one carboxylate group. This effect is not observed, showing that the number of salt-bridges is, in simulations of both folded and unfolded states, preserved among the variants. The absence of the Asp 104 carboxylate group is compensated by other groups as far as salt-bridges are concerned.

An example of this process is displayed by following one pair of opposite charges in the WT sequence: the carboxylate group of Asp 104 on one side and the guanidinium group of Arg 97 on the other side. To better guide in the following analysis, in Figure 8 we display folded (left) and unfolded (right) representative structures for WT (top) and D104G (bottom) variant. The residues and motifs studied in the following are displayed.

In WT we measured the distance between Cγ (Asp 104) and all other positively charged groups in the protein. By inspecting the sampled distance in the meta-statistics, we found that only one pair has probability larger than 0.1 to form a salt-bridge: Nη(Arg 97). The distribution of the Cγ(Asp 104)-Nη(Arg 97) distance is plotted in Figure 9 (top panel) in the folded (black curve) and unfolded (red curve) states. The salt-bridge is formed in the unfolded state, because in the majority of the configurations represented by the folded state, Asp 104 side-chain is projected towards the solvent by the α1 orientation forced by the β-sheet scaffold. Since the β-sheet is demolished in the unfolded state, the Asp 104 side-chain becomes available to form intra-molecular salt-bridges and Arg 97 is the most suitable partner.

Arg 97 suitability is shown by analyzing the probability, in the meta-statistics, of salt-bridges formed by its guanidinium group. In meta-statistics, Arg 97 is involved for most of the configurations in salt-bridges with Glu 100 and Glu 101, beside Asp 104. These observations show that Arg 97 is involved in neutralizing the negatively charged patch located in α1 helix, to which Glu 100, Glu 101, and Asp 104 all belong. The distribution of distance pairs involved in this neutralization is displayed in Figure 9 (middle panel). It can be noticed that in folded configurations Glu100-Glu101 is efficiently neutralizing Arg 97, while in unfolded configurations the neutralization is made by Asp 104 (top panel).

When Asp 104 is replaced by a small residue like Gly, the extent of neutralization of Arg 97 by Glu100-101 is stronger, both in folded and unfolded states (Figure 9, bottom panel). The comparison of the shape of the Arg 97-Glu 100-101 distribution between D104G variant (bottom panel) and WT (middle panel) shows a tighter network of electrostatic interactions in unfolded states (red curve) when Asp 104 is replaced by Gly.

To better understand the effects of the release of the α helices upon unfolding the β-sheet, we studied the relative orientation of the α1 and α2 almost rigid segments. The movement of α1 and α2 helices was analyzed by monitoring the angle between two unit vectors approximately describing the directions of the two helices. We chose the unit vectors along directions Cα(94)-Cα(112) and Cα(184)-Cα(191) for α1 and α2, respectively. The angle between the two unit vectors, $\beta_{1,2}$, describes approximately the angle between the long axes of the two helices. In Figure 10 the distribution of $\beta_{1,2}$ in folded and unfolded states for WT (top panel) and two representative variants, A107V (middle panel) and D104G (bottom panel). The distribution of $\beta_{1,2}$ in the folded state displays a high peak at about 15${}^{\circ }$, with a shoulder at about 45${}^{\circ }$, the latter absent for D104G variant. The low population of $\beta_{1,2}$ at the shoulder in the folded state for WT and A107V denotes a significant plasticity of the helical segments, that is hindered by the replacement of Asp 104 with Gly (D104G variant). D104G displays a barrier for the transition in $\beta_{1,2}$ also in the unfolded state, while the same barrier does not appear both for WT and A107V. The rigidity in the change of relative orientation between the two α-helices in D104G is, therefore, related to the interactions within each segment (see electrostatic interactions displayed in Figure 9) and between the two helical segments. The α1 compaction due to replacement of Asp 104 with Gly increases the chance to keep the hydrophobic patch between the two helices shown by the crystal structure.

### 2.5. Side-Chain and Main-Chain Hydration

Following the analysis described above, we analyze here the change of solvent accessibility of different portions of the FXN protein along the unfolding process. First we computed the SASA of the backbone and side-chain atoms for each residue in the sequence. In Appendix A the distribution of SASA summed over different groups of atoms is displayed after imposing the average values of CV according to folded (14, black curves) and unfolded (2, red curves) states. We divided the SASA into six groups: Backbone and side-chain atoms for all residues (90-210, left panels); backbone and side-chain atoms in α1 motif (residues 92-114, central panels); backbone and side-chain atoms in α2 motif (residues 182-193, right panels). We first describe the WT sequence (top panels of the figure). The backbone of the entire protein, as expected, displays a significant increase in SASA, because by breaking hydrogen bonds involving backbone atoms the peptide linkages become more exposed to the solvent. By inspecting the contributions of α1 and α2 we notice that the first motif is affected to an extent similar to the total backbone, while the second motif is not, especially in the backbone. These data confirm that by unfolding the β scaffold, α1, in the N-terminus, is more destabilized than α2, in the C-terminus. This effect is also shown by side-chains. Since the number of side-chain atoms is larger than the number of backbone atoms, the change in total SASA for the side-chain groups looks larger than that of backbone groups. To compare SASA for different groups of atoms, it is convenient to divide the SASA of each group by the maximal value found in the meta-statistics for the same group of atoms. We call this quantity the relative SASA. The change of relative SASA for side-chain atoms is smaller than that of backbone (data not shown here for clarity). During the unfolding process, side-chains tend to be kept stuck one to each other, while the backbone atoms are more efficiently soaked by water molecules. Because of the differences in absolute SASA between side-chains and backbone, however, the contribution to energy changes is larger for side-chains than for backbone.

The comparison between WT and the studied variants (the other rows in the same figure, Appendix A) is useful to understand the peculiar behavior of D104G compared to the other variants. The first observation is that the black curves for WT and D104G are narrower than for most of the other variants. The D104G variant displays a wider change of α2 side-chains with unfolding, showing that α2 is accommodating the changes in the protein scaffold better than in most of the other cases, that are generally less perturbed by the unfolding process. It must be noticed that, in the folded state, D104G has the lowest SASA for side-chain atoms in α1, being the maximum of the peak in the black curve of middle panels significantly shifted compared to all other variants. Most of this effect is due to the removal of Asp 104 side-chain in D104G. Glycine residues are, for all variants except D104G, located in the β1–4 scaffold, where they are efficient in stabilizing the β-strands. However, Gly 104 in D104G is very efficient both in reducing the size of α1 and in constraining the α1 motif to its helical configuration.

It is interesting to focus on α1 motif and on Arg 97 because the latter residue represents one of those residues with a net charge belonging to α1 motif and shared among all variants. The relative SASA of backbone atoms in Arg 97 is very small in the folded state for all variants, consistently with the involvement of Arg 97 in the α1 helix. When the protein is unfolded, the average relative backbone SASA of Arg 97 increases, while the average relative SASA of side-chain atoms of Arg 97 decreases. This occurs because the Arg 97 side-chain is never fully projected into the solvent. In both folded and unfolded states, Arg 97 side-chain is trapped into a network of salt-bridges (see also Figure 9) and intra-molecular hydrogen bonds. Within this network, electrostatic interactions and the polar contribution to the effective energy of Equation (Equation 12) play an important role along most of the unfolding pathways.

In the frame described above, the D104G variant displays a rigid and compact α1 motif, because of the small Gly 104 side-chain that stabilizes the α-helix containing most of the charged side-chains. In the D104G variant the rigidity of the α1 motif and its lower net charge hinders, in the free energy change with respect to WT, the accommodations that occur in the other variants.

To address the eventual displacements of C- and N-termini in FXN, we analyzed the position of the C-terminus in the two reference structures of FXN, 1EKG and 5KZ5. For instance, the distance between N of Leu 90 and C of Ala 210 is 28.0 and 18.4 Å in 1EKG and 5KZ5, respectively. This indicates that the C-terminus comes closer to residue 90 when the structure has the shape suitable to fit into the mitochondrial complex. In our simulations we observe that this distance is distributed among many peaks, both in folded and unfolded states. In Figure 11 the distribution of such distance is plotted for the WT sequence, being the peak positions in folded and unfolded structures similar for all variants. The salt-bridge represented by the first sharp peak accounts, in the folded state, for less than 5% of the configurations. This salt-bridge is responsible of a relatively rare protein sealing. The two peaks at 18 and 24 Å contain most of the configurations close to the initial configurations chosen in the 90 walkers, that are close, respectively, to 5KZ5 and 1EKG. About 39% of the configurations contribute to the peaks within a distance of 20 Å while the percentage decreases to 27% in the unfolded state. However, we notice a significant increase in population in the region 4–10 Å in the unfolded state, indicating that once the salt-bridge is broken the C-terminus becomes trapped into electrostatic interactions hindering the complete release of the weak protein sealing. This behavior indicates a little resistance of C-terminus to move away from the N-terminus and, therefore, from α1 region.

## 3. Discussion

The comparison between the calculation of free energy changes of variants with respect to that of the WT sequence (Table 3) shows that the method here applied is able to capture the effects of interactions either hindering or assisting the unfolding of the tertiary structure of FXN. As for the different extents of destabilization, the method is not accurate. Therefore, we propose the method to predict the effect of single-point mutations on the sign of free energy changes. A wider set of stabilizing single-point mutations has been recently published [13]. We did not predict the behavior of the latter variants because, differently from the 8 variants studied here, they are not associated to cancer, that is the goal of our study. These variants will be part of a second study.

The low prediction ability of the present method as for the absolute value of free energy changes is common to all other methods reported in the literature (see Table 3). As for atomistic models, the origin of the deviation from experiments is related to both the approximation of the force-field and to the limits in sample size and statistics. The method here proposed enhances the statistical accuracy, especially in the unfolded state. Compared to other atomistic methods reported in the literature [12,13,15,18], that sample the deformation of the native state at room conditions, the sampling forced by collective variable we have chosen allows the inclusion in the statistics also of configurations far from the native state. Therefore, the possible contributions of these configurations to protein plasticity and to functional switch are more appropriately taken into account.

The function of FXN is both to store Fe ions in a disposable way and to assist the transport of Fe ions from the storage to the FeS cluster assembly machinery in mitochondria [11,25]. Thanks to advancements in crystallography and cryo-EM experiments, structures are available for forms suitable to storage [26] and FeS assembly [23]. The FeS-assembly complex is formed by proper interfaces of FXN with ISCU and NFS1 proteins [23,27]. Even though the Fe storage mechanism is debated, this function is likely performed via self-assembly of FXN into oligomers [26,28]. The structure of FXN oligomers have been determined for yeast (PDB 4EC2). Since the structural similarity of 4EC2 and 1EKG is large [11], we assume that 1EKG represents a monomer structure suitable to fit into experimentally available oligomeric structures. Thus, the larger the similarity of a configuration to 1EKG, the larger is the chance for such configurations to adapt into oligomeric assemblies suitable to store Fe.

Any structural similarity to 5KZ5 is, on the other hand, an indication of a large chance to fit into the active mitochondrial complex. We remind that RMSD of C, O, N, and Cα atoms in segment 90-210 between 1EKG and 5KZ5 is about 5 Å: therefore, the two structures are different. A smooth transition from monomers adapted to self-assembled oligomers towards configurations adapted to the mitochondrial complex is an indication of FXN plasticity, allowing a smooth change of function induced, respectively, by the presence in the FXN environment of a large number of identical partners (storage) or the presence of the proteins of the mitochondrial complex (NSF1, ISCU, and other cofactors). Moreover, changes in Fe concentration can affect the plasticity of FXN.

By comparing the collected statistics we simulated to both these reference structures, 1EKG and 5KZ5, we obtained that the D104G variant is more suited to the functional switch required by the Fe-S cluster assembly process. This variant is constrained to minimal structural changes with respect to both reference structures. The interactions responsible of this constraint are concentrated in the α1 motif. First, the α1 motif is kept rigid by the Asp→Gly mutation. Then, the negative charge in α1 is reduced while keeping efficient the neutralization of negatively charged residues in α1.

Conversely, all other variants are, both computationally and experimentally, more easily unfolded in the region 90-210. The computational model shows that the electrostatic repulsion between those residues concentrated in the α1 helix is more easily dissipated by movements of α1 helix in the crowded environment provided by the unfolded β-sheet (Table 2 and Table 4). In these cases, therefore, any pathological effect of FXN sequence variation can be associated to a loss of both interfaces: that associated to self-assembly (storage) and that associated to the assembly into the active mitochondrial complex (Fe-S biosynthesis).

The question why D104G single-point mutation has the clinical implication of liver carcinoma is beyond the possibility of the here reported model. However, the model provided useful working hypotheses. One possibility is that removal of one Asp side-chain reduces the possibility of a strong Fe-binding, potentially exerted by residues with a carboxylate group in the side-chain, located in negatively charged protein patches [29], and projected into the solvent. Even though the most stable Fe-binding site is assisted by His 86, that is missing in the sequence we simulated, Asp 104 is also affected in experiments by Co^2+^ addition [30]. The Fe-binding to FXN 90-210 was also confirmed by recent EPR and NMR measurements [31], thus showing that His 86 is not essential to Fe binding. The addition of Fe^2+^ to FXN 90-210 produces the largest amide N chemical shift change in Asp 104 compared to all other residues.

The change of Asp 104 into a small amino acid, as in the D104G FXN variant, has the consequence of a larger protein stability, but a lower Fe-binding propensity.

The role of the C-terminus in the stability of FXN folding has been addressed by NMR, fluorescence and CD experiments where the sequences 90-210 and 90-195 have been compared [12,32]. These experiments were found consistent with the presence of at least two sub-states pointing to a local unfolding of the C-terminal region before a global unfolding occurs. We analyzed the C-terminus position using the N(Leu 90)-C(Ala 210) distance as a useful structural parameter. Despite a wide movement of the C-terminus away from the N-terminus upon global unfolding, the pathway of the transition is hindered in the WT sequence. This observation is an indication of possible partially unfolded states close to the native structure. The statistical weight of such configurations, however, is small compared to configurations displaying large end-to-end distance. Therefore, these transient states are not expected to affect experimental observations like NMR cross-peaks.

## 4. Conclusions

We applied a recently developed computational method to sample, with reasonable computational resources, molecular statistics including folded and unfolded states for a truncated sequence of human frataxin (FXN), the 90-210 isoform. This sequence has been extensively studied both in vitro and in vivo. We were able to interpret the measured changes in thermal stability of native FXN and eight variants related to different pathologies. The computed changes of free energy upon general FXN unfolding show that one variant (D104G) is more resistant to unfolding compared to all other variants investigated here, consistently with thermodynamic experiments. Experimentally, all of the variants that are somatic mutations inducing cancer, display a propensity to unfolding larger than native sequence, with the exception of D104G.

The different behavior of D104G variant, compared to all other here examined single-point mutations, shows that the pathology associated to D104G variant, liver carcinoma, is likely associated to the specific features of Fe-binding, while the pathologies associated to all other mutations deal with the lower thermal stability of FXN.

The method proposed in this work is effective in predicting whether a single-point mutation enhances or decreases the stability of a given protein scaffold. Future applications of the method are for the same protein in the presence of a protein partner, like iron ions or other proteins involved in the mitochondrial complex.

The method can be applied to many biological macromolecules to characterize their transition from experimentally defined folded states to unfolded states that keep different degrees of structural information. The studied transition is strongly biased by the chosen collective variable. Strategies to correctly choose and describe the collective variables have been proposed in the field of metadynamics. Any of these strategies can be fruitfully used in combination with the method here proposed.

## 5. Methods

### 5.1. Metadynamics

Let ξ(q) be a collective variable (CV) that is a function of atomic positions, *q*. When ξ is an observable quantity, the values, *s*, allowed for ξ can be used to label system macrostates, while the set of coordinates *q* labels the system microstates, each set of *q* yielding one of the possible values of *s*. If ergodicity holds, infinitely long simulations of a trajectory q(t) in a given statistical *ensemble* would correctly sample the statistical weight of ξ. However, because of the huge number of ways in which certain values of *s* of ξ are encountered, compared to others, actual numerical simulations in practice only sample the maximally degenerate values of ξ.

Standard statistical *ensembles* and more recently generalized *ensembles* try to address this problem by biasing the trajectory to spend more time where ξ has a low degeneracy and less time where ξ has a large degeneracy. Actually, since these statistical *ensembles* are sampled at a given temperature *T*, the bias affects the probability of ξ, rather than its degeneracy.

The sampling of configurations obtained with the biased inverse probability of ξ is called metastatistics. We will denote by $\tilde{P}\left(q\right)$ the probability of microstates encountered along the simulated trajectory and by $\tilde{P}\left(\xi \right)$ the probability of the macrostates labeled by ξ. For simplicity with a little abuse of notation we use the same name for the metastatistics probability as function of the microscopic variables, *q*, and to the associated metastatistics probability as function of the macroscopic collective variable, ξ.

Many methods have been proposed to sample configurations with the inverse of the estimated probability of ξ [33]. In this work, we used the altruistic metadynamics proposed in Ref. [34]. The desired metastatistics is obtained from a swarm of trajectories provided by metadynamics after building a suitable external bias that is then kept fixed when collecting configurations in the final step of the NpT simulation (see Section 5.6). We used simulations in the statistical ensemble associated to constant temperature *T* and pressure *p* (NpT ensemble) because macromolecules undergoing large conformational changes when experiencing an external bias exert strong perturbations over the explicit solvent and ions representing the environment of the macromolecule. To cope with steep changes of kinetic energy of water molecules and possible temporary voids around the macromolecule, the NpT ensemble is recommended.

In the framework of metadynamics, the estimated probability of the CV is expressed by means of a sum of gaussian functions, $V_{G}\left[\xi \left(q\right)\right]$, related to the metastatistics probability by the formula
(1)$$\ln\tilde{P}\left(\xi \right)=\beta V_{G}\left[\xi \left(q\right)\right]+C,$$
with $\beta =1/(k_{B}T)$ where *T* is the temperature used in the simulation, $k_{B}$ the Boltzmann constant, and *C* a normalization constant that is of no relevance in the computation of thermal averages. Different methods have been proposed to build an external bias $V_{G}\left[\xi \left(q\right)\right]$ such that the distribution of ξ is flat and transitions between folded and unfolded states of a biomolecule endowed with many degrees of freedom, are equally well sampled.

### 5.2. The Maximal Constrained Entropy Method

Despite the fact that there exist dynamical methods (metadynamics) by which one can collect large sets of configurations (metastatistics), it is a fact that the sampled configurations can only provide an approximate (but not quite) flat distribution of the CV. To circumvent this difficulty we propose to use the maximal constrained entropy method to construct, starting from $\tilde{P}\left(\xi \right)$, a better probability for thermal average calculations.

Since in actual simulations one works with trajectories where configurations can be enumerated, we attach the microstate index γ to the configuration *q* and we denote by ${\tilde{P}}_{\gamma }$ the probability
(2)$${\tilde{P}}_{\gamma }=\frac{{\tilde{w}}_{\gamma }}{\sum_{\gamma }{\tilde{w}}_{\gamma }},$$
where ${\tilde{w}}_{\gamma }$ is the number of microstates with label γ collected in the metastatistics and $\tilde{Z}=\sum_{\gamma }{\tilde{w}}_{\gamma }$ is a normalization factor.

In an infinitely long (ergodic) simulation, it is unnecessary to explicitly evaluate the weights ${\tilde{w}}_{\gamma }$, as they are automatically encoded in the degeneracy of the set of collected configurations sampled along the simulated trajectory. This means that in the following equations, where the sum over γ is extended over that actually produced configurations, we should not include the factor ${\tilde{P}}_{\gamma }$. However, we leave this redundant factor to recall that we are dealing with the set of configurations generated by metadynamics. As we said, the resulting metastatistics is never exact as the CV distribution does not come out to be flat. Thus we ask how we can at best exploit the information contained in the collected configurations to compute thermal averages and in particular how we can deal with the problem of computing the expectation value of the free energy as a function of the values taken by the chosen CV. A viable solution of this problem is provided by the maximal constrained entropy method.

Given an estimate, ${\tilde{P}}_{\gamma }$, of the metastatistic probability, say the one provided by metadynamics, the problem of finding the least-biased expression of the probabilities $P_{\gamma }$ that is nearer to ${\tilde{P}}_{\gamma }$ and satisfies the condition
(3)$$s=\langle \xi \rangle =\sum_{\gamma }P_{\gamma }\xi_{\gamma },$$
is solved by determining the maximum of the cross-entropy functional [21,35,36]
(4)$$S_{c}\left[P,\tilde{P}\right]=-\sum_{\gamma }P_{\gamma }\ln\frac{P_{\gamma }}{{\tilde{P}}_{\gamma }}.$$
under the constraint (Equation 3). The well-known solution of this variational problem is given by the formulae:(5)$$P_{\gamma }=\frac{{\tilde{P}}_{\gamma }}{Z_{\lambda }}\exp\left(-\lambda \xi_{\gamma }\right)$$
(6)$$Z_{\lambda }=\sum_{\gamma }{\tilde{P}}_{\gamma }\exp\left(-\lambda \xi_{\gamma }\right)$$
with the parameter λ the solution of the (highly non-linear) equation:(7)$$s=\sum_{\gamma }P_{\gamma }\xi_{\gamma }=\frac{1}{Z_{\lambda }}\sum_{\gamma }{\tilde{P}}_{\gamma }\exp\left(-\lambda \xi_{\gamma }\right)\xi_{\gamma }.$$

The quantity $\exp\left(-\lambda \xi_{\gamma }\right)/Z_{\lambda }$ is called the modulation factor of the metastatistics, hereafter. Owing to Equation (Equation 7), λ is a function of *s*. If desired, we can thus trade the dependence upon λ for the dependence upon *s* and vice versa.

Plugging the solution for $P_{\gamma }$ into Equation (Equation 4) one gets for the cross entropy at its maximum:(8)$${\bar{S}}_{c}\left(s\right)=\ln Z_{\lambda }+\lambda \;s.$$

This quantity is a measure of our ignorance once the (approximated) metastatisics probability ${\tilde{P}}_{\gamma }$ and the average value of the CV, *s*, are given. The average of any other configurational quantity B(q) can be computed then and takes the form:(9)$$b_{\lambda }={\langle B\rangle }_{\lambda }=\frac{1}{Z_{\lambda }}\;\sum_{\gamma }{\tilde{P}}_{\gamma }\exp\left(-\lambda \xi_{\gamma }\right)\;B\left(q_{\gamma }\right),$$
with γ running over the configurations (microstates) collected along the metadynamics, and $q_{\gamma }$ the atomic configuration of each microstate. This equation holds also for the quantity which is of main interest here, namely ${\langle H\rangle }_{\lambda }$ with H=U+pV the enthalpy of the system. With obvious notations *U* is the potential energy, *p* the constant pressure in the NpT ensemble and *V* the volume of the system (that we recall fluctuates in NpT simulations).

### 5.3. Estimating the Free Energy

The central goal of our work is to estimate the change in free energy upon introducing certain information (in our case the knowledge about the degree of protein folding) into the metastatistics, while keeping maximal our ignorance about all other parameters. As in our previous work [36], we use as a workable definition of free energy the formula:(10)$$F\left(s\right)={\langle H\rangle }_{\lambda }-T\;k_{B}\;{\bar{S}}_{c}\left(s\right),$$
which is given by the combined sum of the enthalpy in the NpT ensemble (or of the energy *U* if one is working in the NVT ensemble), and the (informational) entropy measured by the maximal cross-entropy. As we said, in Equation (Equation 10) H=U+pV is the enthalpy of the simulated system, λ the parameter associated with the constraint 〈ξ〉=s, ${\bar{S}}_{c}$ the maximal cross-entropy change due to the introduction of such a constraint, $k_{B}$ the Boltzmann constant, and *T* some temperature in the stability range of the system under study. We remind that the maximal cross-entropy is given by Equation (Equation 8) and the average of *H* (or simply of *U* in NVT simulations) is obtained using Equation (Equation 9).

It should be noted that the free energy computed by means of Equation (Equation 10) is not the free energy change usually defined in metadynamics. The latter is given by:(11)$$F\left(s\right)-F\left(s_{0}\right)=-\left[V_{G}\left(s\right)-V_{G}\left(s_{0}\right)\right]$$
with $s_{0}$ a reference state for the collective variable and $V_{G}$ the external biasing potential determined in such a way that the same atomistic system is discouraged from passing over already visited configurations. In an ideal simulation, after an infinitely long trajectory, in these circumstances one would end with a flat histogram for the values taken by the CV. Since the actual resulting metastatistics does not yield a flat histogram for the CV, Equation (Equation 11) is not valid and we had to make recourse to the maximal constrained entropy method in order to construct the optimal estimate of the probability, $P_{\gamma }$, starting from the metastatistics probability, ${\tilde{P}}_{\gamma }$.

The direct calculation of ${\langle H\rangle }_{\lambda }$ in Equation (Equation 10) includes the effects of the fluctuations of *U* and *V* due to the movement of all explicit water molecules and ions included in the atomistic model of the protein environment. As it is customary in these cases, we also produce an approximate evaluation of ${\langle H\rangle }_{\lambda }$, where *H* is replaced by the effective mean-field free energy $\bar{U}$ of the protein solute. The nice thing about this approximation is that the energy of the system is still thermally averaged over the many degrees of freedom of water molecules and ions surrounding the much smaller solute protein aggregate.

A widely used strategy for the evaluation of the effective mean-field energy of the solute protein is the so-called molecular mechanics/Poisson–Boltzmann solvent accessible approximation (MM/PBSA) [37]. In this approach the mean-field energy for solute-solvent interactions is described as the sum of polar (electrostatic) and non-polar (surface) contributions. For each protein configuration *Q* one writes:(12)$$\bar{U}\left(Q\right)=U_{intra}\left(Q\right)+U_{solv,np}\left(Q\right)+U_{solv,pol}\left(Q\right),$$
where $U_{intra}$ is the intra-molecular part of the potential energy in the protein force-field, given by
(13)$$U_{intra}\left(Q\right)=U_{str}\left(Q\right)+U_{bend}\left(Q\right)+U_{tors}\left(Q\right)+U_{vdw}\left(Q\right)+U_{el}\left(Q\right).$$

The various contributions are the stretching ($U_{str}$), bending ($U_{bend}$), and torsional ($U_{tors}$) terms in the potential. $U_{vdw}$ and $U_{el}$ are the Lennard–Jones and Coulomb interactions, respectively, computed by summing over all the pairs of atoms of the protein.

The last two terms in Equation (Equation 13) represent solute-solvent contributions and are approximately computed as follows. The term $U_{solv,np}$ is the contribution to the solute-solvent free energy due to the formation of a cavity of zero charge density with the shape of the solute protein and the creation of the solute-solvent interface. Introducing a charge density in the space occupied by the solute leads to the $U_{solv,pol}$ contribution. The charge density is given in terms of the point charges $q_{i}$ of the atom *i* sitting at the point ${\vec{r}}_{i}$, where *i* runs over the $N_{a}$ atoms of the solute molecule.

The term $U_{solv,np}$ is calculated as an empirical linear combination of the solvent accessible surface area (SASA) for each group in the solute molecule [38] according to the formula:(14)$$U_{solv,np}=\sum_{i}^{N_{a}}\sigma_{i}{\mathrm{SASA}}_{i},$$
where the coefficients $\sigma_{i}$ are positive or negative for hydrophobic or hydrophilic groups, respectively, (see below for details). Finally the electrostatic contribution to the solute-solvent free energy, $U_{solv,pol}$, is given by the electrostatic energy required to charge the low-dielectric solute molecule of generic shape into a high-dielectric medium like a salt-water solution. This contribution is obtained by a numerical finite difference solution of the Poisson–Boltzmann equation [39].

### 5.4. Estimating Error on Free Energy

Estimating the error about the main target of this work, that is the free energy change upon unfolding, is difficult, because of the simulataneous presence of many error sources. One major error source is due to slow convergence of the external bias during its construction. This error is discussed in Ref. [40]. The second major error source is in the propagation to averages of the error on λ parameter. This error is also due to statistical limitations on the biased trajectory that is used in the maximal constrained entropy procedure. We focus our attention on the error propagation from λ on the evaluation of the quantity on the lefthand side in Equation (Equation 10). We remark that this equation contains two terms in the righthand side that are computed, respectively, with Equations (Equation 8) and (Equation 9). The latter two equations contain the parameter λ. Since fluctuations of energy contributions in the first addend in Equation (Equation 10) due to solvent and ions degrees of freedom are large in samples with a small number of atoms, we estimate error on *F* by using Equation (Equation 12) in place of enthalpy *H*.

Finally, as usual, error is computed by dividing the biased trajectory into blocks [41]. The error on computed averages is measured as the root-mean square deviation of averages computed over the different blocks.

### 5.5. Summary of the Method

The computational protocol we have presented in this theoretical analysis can be summarized as follows. One starts by performing MD simulations at T=300 K in the presence of the biasing potential $V_{G}\left(\xi \right)$ built according to the metadynamics strategy (Equation (Equation 1)). The resulting statistics is what we call metastatistics. Using the set of collected configurations, we determine the λ parameter that maximizes the cross-entropy $S_{c}$ in Equation (Equation 4) under the constraint 〈ξ〉=s given in Equation (Equation 3). In the present instance, where ξ is the number of hydrogen bonds holding together the protein β sheet secondary structure, *s* takes integer values in the range 0÷15. For each value of *s*, we get a value of λ that once inserted in Equation (Equation 7), yields to the modulation weight:(15)$$w\left[q\left(t\right)\right]=\frac{1}{Z_{\lambda }}\;\exp\left\{-\lambda \xi \left[q\left(t\right)\right]\right\},$$
with *q* the system configuration at time *t*, indexing the microstate γ, along the collected metadynamics trajectory.

### 5.6. Simulation Parameters

In this section we discuss the simulation procedure we followed to compute the expectation values of the physical quantities of interest described in Section 5. In particular, we will provide a brief overview of the altruistic metadynamics method [22,42] we have employed to construct the desired statistical ensemble of configurations.

The main idea of metadynamics [43] is to adaptively bias the system force-field by progressively filling, along the unfolding MD trajectory, the wells of the free-energy surface (FES) profile, F(ξ), viewed as a function of some (judiciously chosen) order parameter, ξ, also called collective coordinate or collective variable (CV). This is done by adding to the potential energy of the atomic system, U(q), an extra low-dimensional (history-dependent) potential term, $V_{G}\left(\xi \left(q\right)\right)$—usually a sum of gaussian functions—progressively constructed in such a way that the system, in its dynamical wandering, is “discouraged” to come back to already visited regions of phase space.

To improve the quality of the phase space exploration we have run in parallel the time evolution of a number of copies of the system of interest (called “walkers” in the following) according to the metadynamics method in its altruistic variant [22,42].

### 5.7. Preparation of Walkers

The initial configurations of the various walkers are obtained from the available crystallographic information about the wild-type FXN protein sequence. In order to assign different initial conditions to the walkers, we used two different structures: The X-ray structure of the mature human frataxin (PDB 1EKG, segment 88-210) [10] and the structure of FXN (PDB 5KZ5, chain A, segment 42-210) in the mitochondrial iron–sulfur cluster assembly machine as it was determined by electron microscopy (EM) [23]. Both structures are determined in the solid state, but 5KZ5 is characterized by a lower resolution compared to 1EKG. In both cases the segment 90-210 was extracted from the available coordinates, because we aim at comparing calculations to the thermodynamic experiments that were collected for the 90-210 FXN protein segment in ref. [15]. The residues 209 and 210, that are missing in 1EKG, were added and built according to standard all-trans amino acid geometry.

The 1EKG structure was used as initial configuration to set-up a first bunch of 60 walkers, while the 5KZ5 structure was used to construct a second bunch of 30 walkers, for a total number of 90 walkers. Each walker was placed in the simulation box after a random rotation with the center of geometry placed in the center of an orthogonal cell with sides large enough to place periodic images of the protein at 40 Åİn this way we avoid direct pair interactions between nearest images of the solute protein, even when proteins are unfolded.

Owing to these random rotations, the 90 simulation cells have slightly different sides. Each cell was filled with water molecules modelled as in the TIP3P approximation [44], merging the solute protein into a snapshot of MD simulation of TIP3P water at room conditions. The minimal water-protein distance was 1.2 Å. The random rotation imposes, therefore, for each walker different initial forces over the protein atoms that are exposed to the water solvent.

A necessary condition to obtain consistent free energy estimates from the maximal constrained entropy method, is to perform averages over configurations with the same number of atoms and atomic types. Different protein variants have different number of atoms and atomic types. Therefore, the external bias was separately built for each protein variant, by using a number of walkers. However, each protein variant has a different number of water molecules in its environment, because of the different simulation cells built around the initial configuration of each walker. To prepare different walkers with the same number of atoms for each protein variant, the number of water molecules needed to be adjusted. This was done according to the following procedure. One starts by changing the sides of the simulation cell so as to accommodate in each cell a number of water molecules as near as possible to the average number of water molecules present in the 90 cells, ending with cells that have similar, but not quite identical, number of water molecules. This cell side adjustment was repeated until a deviation of not more than 20 water molecules of the number of water molecules of the 90 cells is attained. The final step of this adjustment procedure was to delete a few more water molecules at the edge of each simulation cell in order to have in every cell a number of water molecules equal to that of the cell with the minimal number of them. Finally, an appropriate number of randomly chosen water molecules were replaced by Na and Cl atoms (both the FXN protein termini are charged) so as to end up with a neutral solution at 0.1 M in NaCl salt. The protein and NaCl were modelled with the CHARMM36 force-field [45]. The insertion of NaCl was performed with the autoionize package of VMD [20].

After energy minimization, first of solvent and NaCl and then of the entire system, initial velocities were randomly assigned to atoms according to a gaussian distribution corresponding to T=50 K. As a result, although all the cells have the same protein structure and number of water molecules, their initial configurations differ in their initial atomic forces and velocities.

### 5.8. Choosing the Collective Variable

FXN is folded in a structure where two α-helices lay over a small β-sheet. This ternary structure can be perturbed by destroying the intra-molecular hydrogen bonds that stabilize the β-sheet. We decided to take as a CV the number of hydrogen bonds characterizing the β-sheet formed by 4 anti-parallel β-strands observed both in 1EKG and 5KZ5. The use of such CV as a way to monitor the structural transitions in the protein was inspired by several previous applications of metadynamics [42]. In all we counted 16 N–H⋯O hydrogen bonds between backbone atoms in the 1–4 β-strands. As an analytic function counting the number of hydrogen bonds we used the so-called coordination number. So the CV will be defined by the formula
(16)$$S_{a}=\sum_{i,j}s_{i,j}$$
(17)$$s_{i,j}=\frac{1-r_{i,j}^{6}}{1-r_{i,j}^{8}}$$
(18)$$r_{i,j}=\left|{\vec{r}}_{i}-{\vec{r}}_{j}\right|/d_{0}$$
where *a* identifies subsets of the protein, $\vec{r}$ are the atomic positions, and *i* and *j* are indices running over the N and O main-chain atoms, respectively, listed in Table 1. The CV we have chosen is $S_{\beta ,1–4}$. Thus, in this case, *i* and *j* run over the N and O atom pairs forming the 16 hydrogen bonds characteristic of the β-sheet observed in the crystal structure. The $d_{0}$ parameter is taken equal to 3.5 Å, to obtain CV = 16 for the initial configurations in all walkers, being this distance suitable to identify hydrogen bonds between N donors and O acceptors in the protein backbone. As for pairs involved in salt-bridges instead of hydrogen bonds, $d_{0}$ is 4 Å and *i* runs over terminal C, Cγ of Asp and Cδ of Glu, while *j* runs over terminal N, Nη of Arg and Nζ of Lys. Therefore, in the latter case all possible pairs are taken into account. This sum is indicated by $S_{SB}$, hereafter.

### 5.9. The Simulation Strategy

The external bias was built with the help of the altruistic metadynamics method [34]. Naturally the reason for having multiple walkers is to be able to sample all the values of CV in the physically accessible range $0<s<s_{max}$ as uniformly as possible.

The biasing potential, $V_{G}$, is progressively builds by summing over gaussian functions of the CV (each gaussian function is taken to have unit height and width). Gaussian functions are deposited every 20 ps along the MD simulation.

The number of points in the CV range was 20 (0<s<20), consistently with the width of the gaussian functions. The maximal value equal to 20 was chosen to accommodate possible structures with a number of hydrogen bonds larger than that occurring in the crystal structure equal to 16. To avoid the complete unfolding of the protein, a harmonic wall potential $U_{w}=\frac{k}{2}{\left(\xi -\xi_{0}\right)}^{2}$ with *k* = 100 kcal/mol was added to the bias when ξ<1.

Each walker was simulated for 10 ns in a thermal bath of explicit water molecules plus Na and Cl atoms, as explained above, at T=300 K and p=1 bar, with an external bias built according to the procedure described below. The NAMD package [46] was used for the MD simulations (versions 2.13 and 2.14), and VMD [20] for part of the analysis and visualization.

The equation of motion with periodic boundary conditions in three dimensions are integrated with a time-step of 2 fs constraining [47] the bond distances involving H atoms. Non-bonding interactions had a cut-off at 11 Å and the smooth particle mesh Ewald algorithm [48] for long-range electrostatic interactions was employed. The standard stochastic thermostat [49] and barostat [50] to keep temperature and pressure, respectively, constant were used. The simulation cell was kept orthorhombic while solving the equations of motion.

To give an idea of the size of the systems we are dealing with we mention that in the typical case of the native protein sequence, the simulated system was composed by 1875 protein atoms, 34 Na, 26 Cl atoms, and 13,926 water molecules. Table 6 reports some of the simulation parameters and conditions we used to evolve the 90 walkers.

In order to try to get a more uniform exploration of the set of possible CV values we made use of a variant of metadynamics, called altruistic metadynamics [22]. It consists in periodically assigning to the walkers a biasing potential obtained as a linear combination of the accumulated biasing potential of all the walkers up to that moment. According to Equation (Equation 3) of Ref. [22] this linear combination depends on two parameters, 0≤α≤1 and w≤1, which need to be suitably chosen. The actual values of α and *w* we used in the successive stages of the simulation are reported in Table 6. Where values of α and *w* are not indicated, it is intended that no such linear combination is formed and each walker independently from the others builds up its own biasing potential.

Ending this methodological section we need to stress that our goal is not that of using metadynamics to build a flat free energy profile as a function of the chosen CV, rather to have walkers “spread” over different values of CV so as to obtain a large and diversified metastatistics, exploring configurations spanning all the accessible CV values, that can be modulated by the maximal constrained entropy method.

### 5.10. Mutated FXN Proteins

Single point mutated proteins are constructed by simply replacing the amino acid side chain at the point of the mutation with the mutated one. This operation was carried out in the configurations of the WT FXN sequence collected at the end of equilibration (stage 1 in Table 6). A standard geometry for the modified side-chain was assumed. Possible resulting clashes of the replaced side chain with nearby solvent or protein atoms are dealt with by energy minimization. Once this is done, the simulation procedure outlined in Table 6 for the mutated protein is followed.

### 5.11. Smoothing the Biasing Potential

At the end of stage 6 (see Table 6), i.e., after simulating each walker for a total of 24 ns, the external bias that will be used in stage 7 is not updated any more and the final metastatistics is collected, storing configurations along the last 10 ns long trajectory.

For numerical convenience the accumulated final biasing potential, $V_{G}\left(\xi \right)$, is smoothly interpolated by a polynomial of fourth order. The interpolation is made by fitting the polynomial coefficients to the values of $V_{G}\left(\xi \right)$ in 15 equally spaced points in the range 1.5÷15.5. Beyond these limits, the bias is continuously extrapolated as a linear function. This linear analytical expression, extended beyond the physical range of the CV, avoids using a harmonic potential at the boundary values in the stage 7 of the simulation. Note that as the biasing potential depends on the detailed protein sequence, the interpolating polynomials are different for different mutants.

The interpolation of the biasing potential has further technical advantages. First of all, it allows an analytical calculation of its contribution to energy and atomic forces. Secondly, in principle, its explicit expression could be exploited to cancel the effect of the bias in computing thermal averages (as suggested in the original method [43,51]), or in the application of the maximal constrained entropy method we propose here. However, we decided not to proceed along this way because a proper cancellation of the bias requires an extremely large statistics for all the protein variants in study, much larger that the set collected in stage 7 which amounts to 900,000 configurations for each of the 9 protein sequences we have simulated. This more fundamental approach will be tried in a following work dealing with a significantly smaller protein.

### 5.12. Solving the Equation for λ

The numerical solution of the problem of maximizing entropy $S_{c}$ in Equation (Equation 4), under the constraints (Equation 3), requires determining the parameter λ that satisfies Equation (Equation 7). This is done by minimization of the error function
(19)$$\chi^{2}=\frac{{\left({\langle \xi \rangle }_{\lambda }-s\right)}^{2}}{\sigma^{2}},$$
where ${\langle \xi \rangle }_{\lambda }$ is the average computed at the current value of λ using Equation (Equation 9), *s* is the target CV value, and $\sigma^{2}$ is the expected variance of the target data. Lacking any better estimate of $\sigma^{2}$ we conveniently take $\sigma^{2}$ = 1. In the search for the minimum of $\chi^{2}$, one needs to know the derivative of $\chi^{2}$ with respect to λ. The latter can be expressed in terms of second-order momenta of the CV ξ. Convergence of the minimization iteration (we used the Levenberg–Marquardt method [52]) is assumed when the relative variation of $\chi^{2}$ between two successive iteration is less than $10^{-4}$.

To give an idea of the present computational effort, we recall that the total number of microstates γ sampled in metastatistics is 10,000 (number of configurations collected for each walker in stage 7) times 90 (the number of walkers) and all this for each protein sequence (WT and eight single-point variants).

The code to perform the task described above will be provided as an open-source documented package and it will be a modification of the general time-series analysis proposed in Ref. [53].

### 5.13. Analysis of Structural Parameters

The deviation between protein structures is measured via the minimal root-mean square deviation (RMSD) [54] of non-hydrogen atoms in the protein backbone (N, C, O, and Cα atoms), for those residues (90-210) that are shared among all forms:(20)$$\mathrm{RMSD}={\left(\frac{1}{N}\sum_{i=1}^{N}m_{i}{\left|{\vec{q}}_{i}-{\vec{q}}_{i,0}\right|}^{2}\right)}^{\frac{1}{2}},$$
with *N* the number of atoms *i* in the chosen atomic set, ${\vec{q}}_{i}$ the position vectors of each atom *i* along with the collected trajectories, $m_{i}$ the mass of each atom *i*, the subscript i,0 indicating the position of atom *i* in the reference structure used for the structural comparison. As usual, the RMSD is minimized by rotating one of the two compared structures after translating both structures as to have the origin of the axes in the center of geometry.

### 5.14. Mean-Field Approximation of the Solvent

As we mentioned before in Section 5.3, in order to compute the quantity $\bar{U}$ defined in Equation (Equation 14), we need to evaluate the SASA of each atom appearing in the sum. This was performed with the help of the “Numerical Surface Calculation” (NSC) code [55] using a density of 122 points per unit sphere. The SASA of each group was calculated employing the radii and the σ parameters of Ref. [38]. Hydrogen atoms are taken to have zero radius, in the sense that they are incorporated within the radius of the non-hydrogen atom they are attached to. The probe radius was chosen to be 0.14 nm, and kept fixed in the whole calculation.

For each protein configuration one needs to solve the Poisson–Boltzmann equation for the electrostatic potential consistently with the charge distribution in the protein merged into the continuum solvent. This is done by discretizing the (orthorhombic) cell box containing the protein, after the explicit water is removed and replaced by the continuum environment. We used a $100^{3}$ lattice with a lattice spacing of 0.0875 nm. The protein was placed at the center of the cell surrounded by at least 10 Å of boundary dielectric in every space direction. Debye boundary conditions were adopted in which the electrostatic potential decreases exponentially at the cell boundaries. The solute-solvent energy is obtained in terms of the interactions between the atomic point charges and the polarization surface charge density placed at the solute–water interface [39,56]. The relative dielectric permittivity of the molecule and of the solvent were 1 and 80, respectively. In the concentration conditions we work with (bulk 1:1 physiological salt concentration $C_{s}=0.1$ M) the linear approximation provides a satisfactorily accurate solution of the Poisson–Boltzmann equation.

### 5.15. Error on Free Energy

We divided the trajectory into 100 blocks, each composed of 9000 points. The points are sampled one over 100 in the trajectory of 900,000 points. Different blocks are obtained by shifting the initial point from 1 to 100. This error is discussed for the wild-type sequence, being about the same for all variants. 

## Figures and Tables

**Figure 1 molecules-27-01955-f001:**
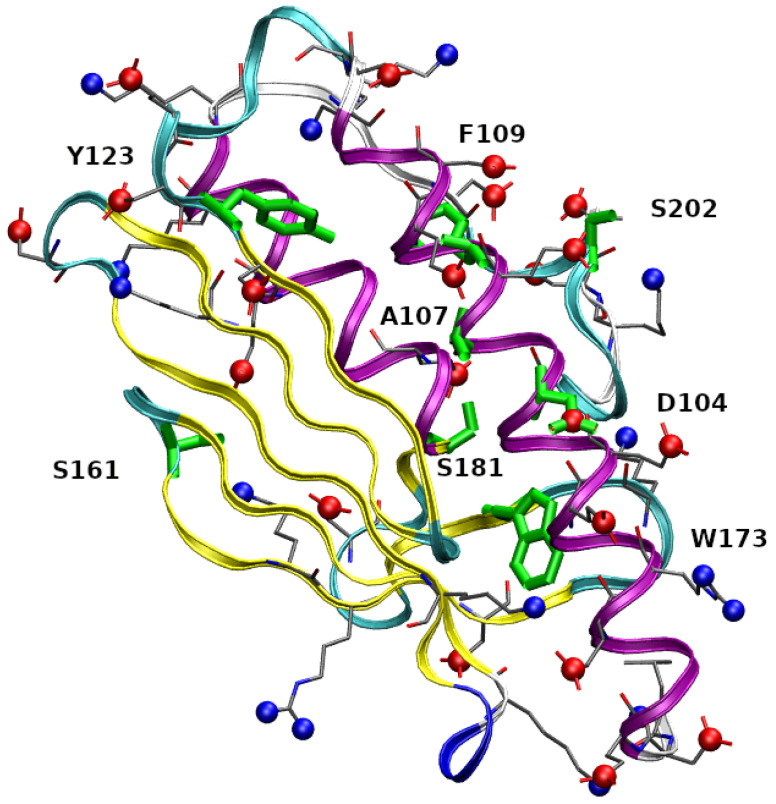
Structure of FXN residues 90-210 in 1EKG crystal. Ribbons interpolate main-chain atoms, with color according to the STRIDE algorithm [19]; β-strands are in yellow; α-helices are purple. Residues that are mutated (with labels) are in green. Red spheres are C in carboxylate groups. Blue spheres are N in ammonium and guanidinium groups. Atomic and bond radii are arbitrary. The VMD program [20] is used for molecular drawings.

**Figure 2 molecules-27-01955-f002:**
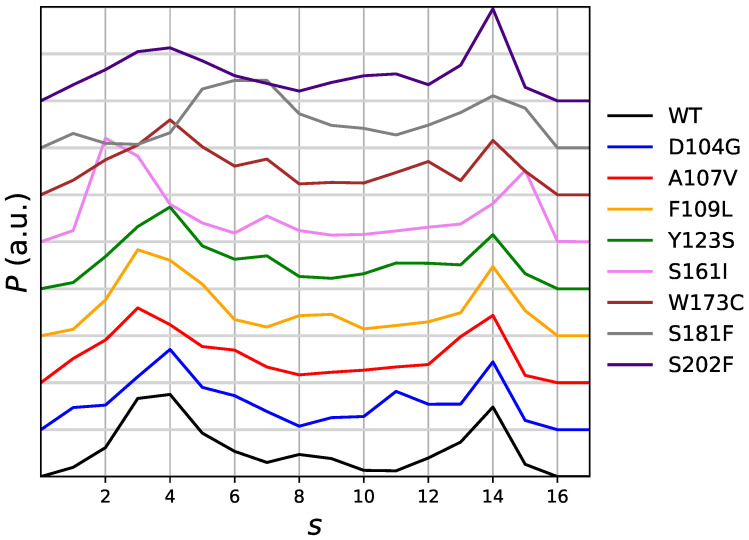
Distribution of collective variable in the meta-statistics (Equation (Equation 2)). Variants are ordered according to increasing mutated residue number. From bottom to top: WT, D104G, A107V, F109L, Y123S, S161I, W173C, S181F, S202F.

**Figure 4 molecules-27-01955-f004:**
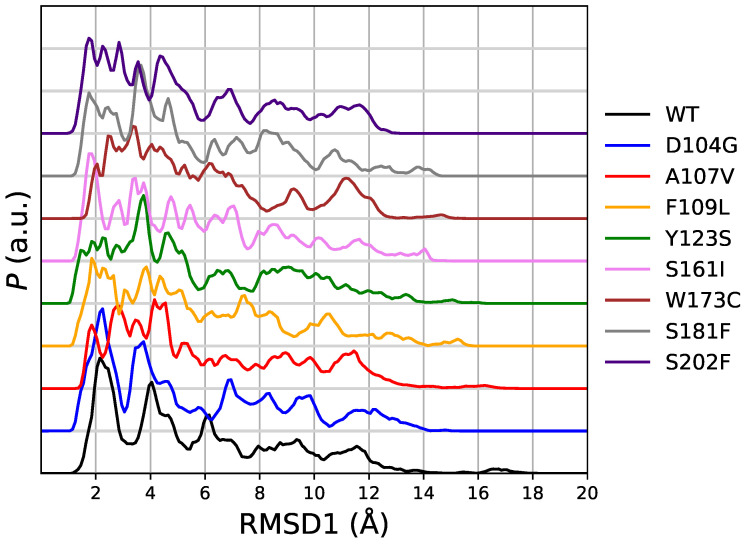
Distribution of root-mean square deviation with respect to PDB 1EKG (RMSD1) obtained in meta-statistics for all studied variants. The order is the same as for Figure 2.

**Figure 5 molecules-27-01955-f005:**
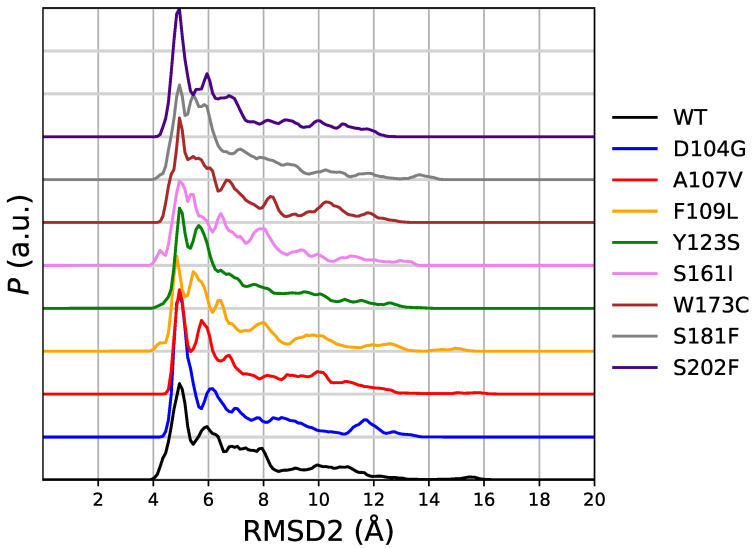
Distribution of root-mean square deviation with respect to PDB 5KZ5 (RMSD2) obtained in meta-statistics for all studied variants. The order is the same as for Figure 2.

**Figure 6 molecules-27-01955-f006:**
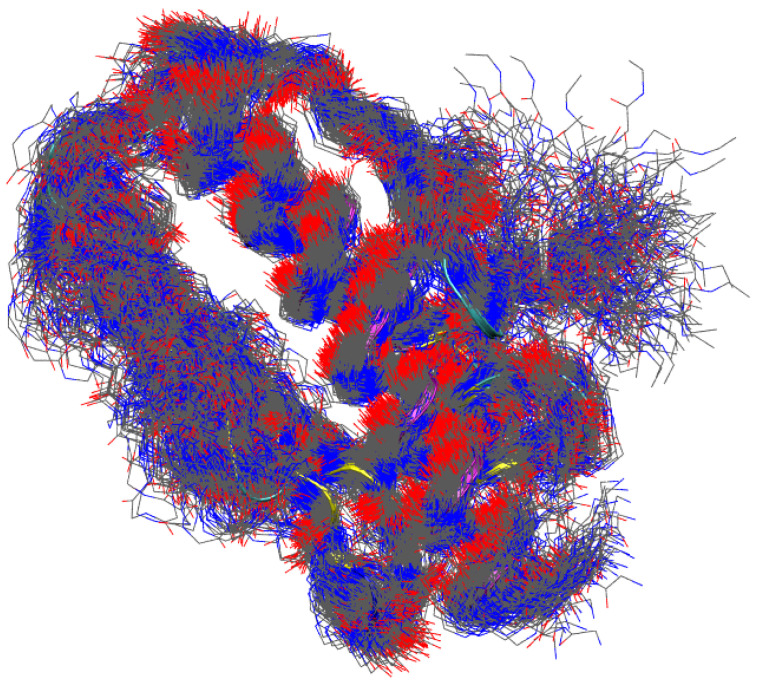
Structures of FXN, WT sequence, with RMSD1 < 3.3 Å. 254 structures uniformly sampled within the 254,298 collected are displayed. Only non-hydrogen backbone atoms are displayed for clarity. The drawing is made in the same protein orientation as in Figure 1. The drawing of 1EKG structure is made using ribbons.

**Figure 7 molecules-27-01955-f007:**
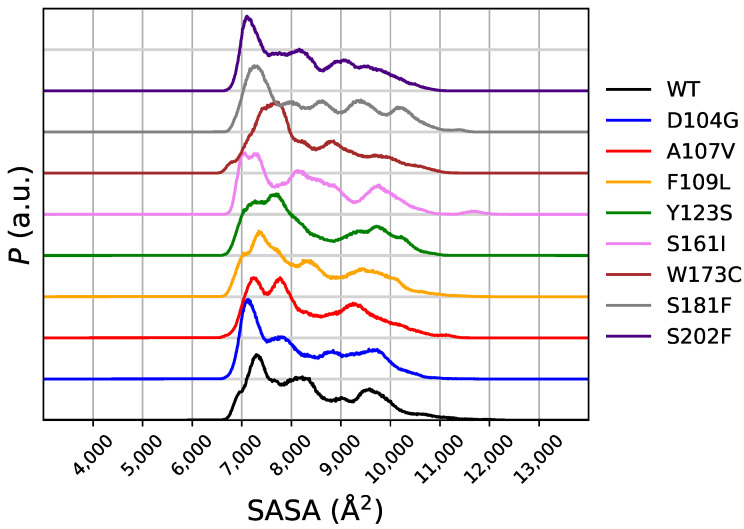
Distribution of solvent-accessible surface area (SASA) obtained in meta-statistics for all studied variants. The order is the same as for Figure 2.

**Figure 8 molecules-27-01955-f008:**
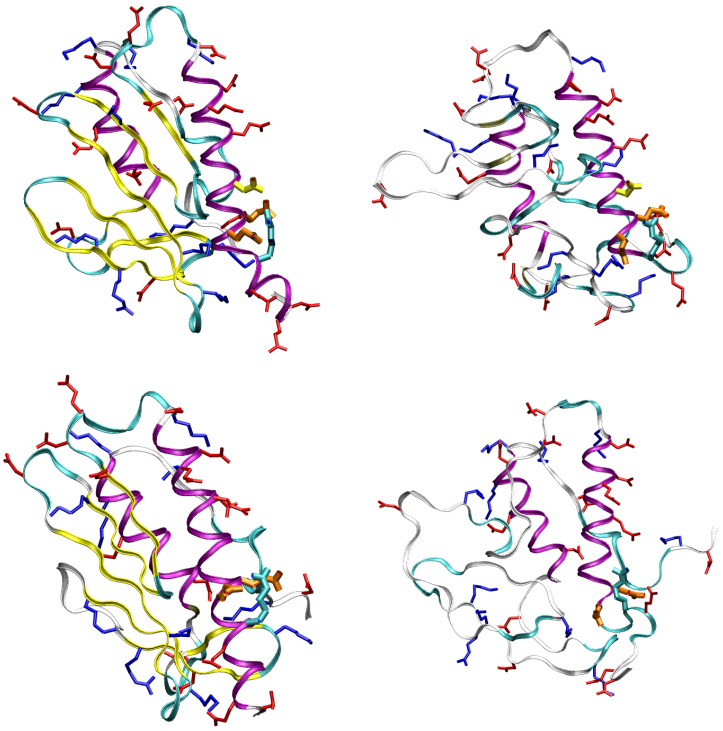
Configurations representing folded (**left**) and unfolded (**right**) states for WT (**top**) and D104G (**bottom**) sequences. Glu 100-101 are represented as orange sticks; Asp 104 (only in WT) is in yellow; Arg 97 is cyano; all other Arg/Lys sidechains are in blue; Asp and Glu sidechains are red. Structural parameters (see also Table 2) are: top-left: $S_{\beta ,1–4}$ = 14, $d_{1}$ < 5 Å, $d_{2}$ < 10 Å, $\beta_{1,2}$ = 5${}^{\circ }$, RMSD1 = 2.7 Å, RMSD2 = 4.3 Å; top-right: $S_{\beta ,1–4}$ = 4, 5 < $d_{1}$ < 10 Å, 10 $<d_{2}<$20 Å, $\beta_{1,2}$ = 30${}^{\circ }$, RMSD1 = 8.4 Å, RMSD2 = 7.5 Å; bottom-left: $S_{\beta ,1–4}$ = 14, $d_{1}$ < 5 Å, 10 < $d_{2}$ < 20 Å, $\beta_{1,2}$ = 21${}^{\circ }$, RMSD1 = 1.6 Å, RMSD2 = 5.0; bottom-right: $S_{\beta ,1–4}$ = 4, 7.5 < $d_{1}$ < 12.5 Å, 10 < $d_{2}$ < 20 Å, $\beta_{1,2}$ = 39${}^{\circ }$, RMSD1 = 6.6 Å, RMSD2 = 6.3 Å. $d_{1}$ is the distance analyzed in Figure 9; $\beta_{1,2}$ is the angle analyzed in Figure 10; $d_{2}$ is the distance analyzed in Figure 11.

**Figure 9 molecules-27-01955-f009:**
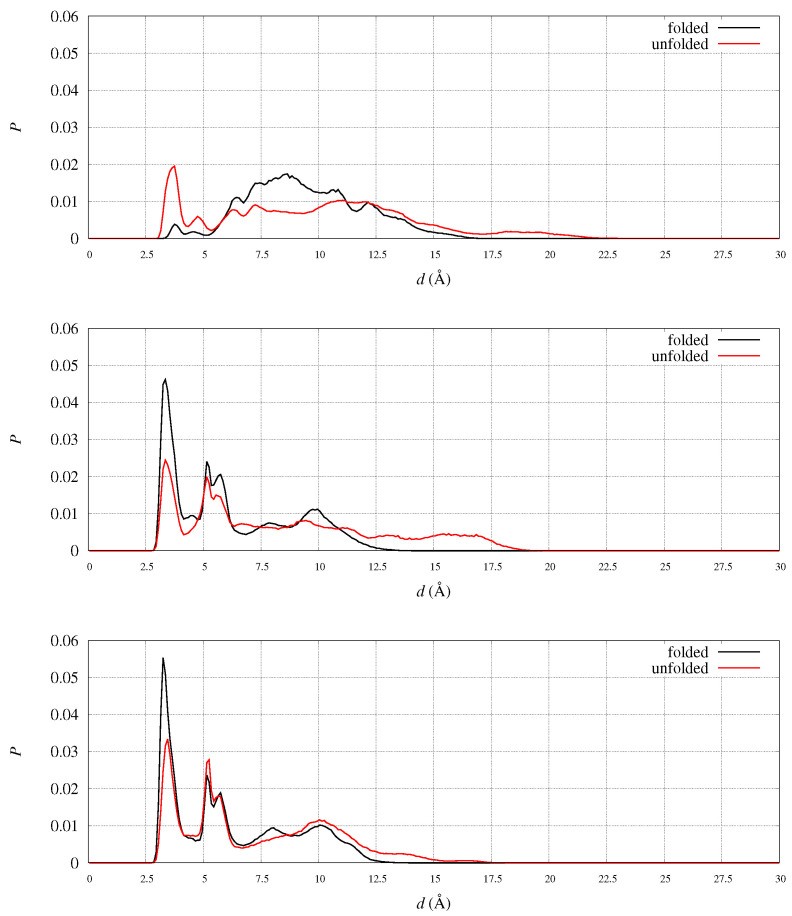
Distribution of selected distances in folded (CV = 14, black curve) and unfolded (CV = 4, red curve) states. Top—*d* is the distance between Nη(Arg 97) and Cγ(Asp 104), WT sequence; middle—*d* is the distance between Nη(Arg 97) and Cδ(Glu 100-101), WT sequence; bottom—same as in the middle, for D104G variant.

**Figure 10 molecules-27-01955-f010:**
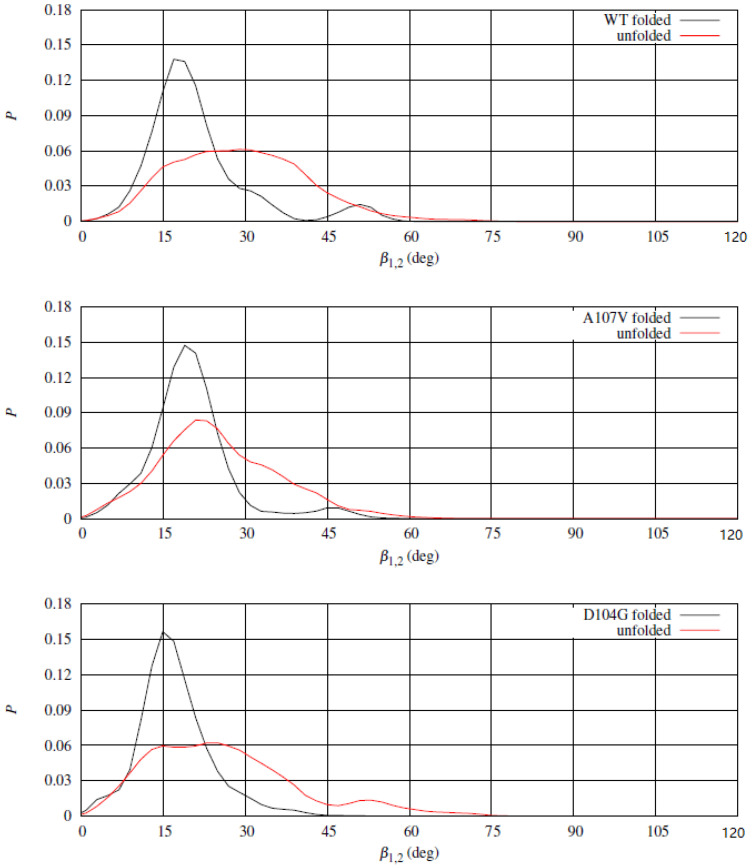
Distribution of the angle between long axes of α1 and α2 segments ($\beta_{1,2}$) in folded (CV = 14, black curve) and unfolded (CV = 4, red curve) states. Top—WT sequence; middle—A107V variant; bottom—D104G variant.

**Figure 11 molecules-27-01955-f011:**
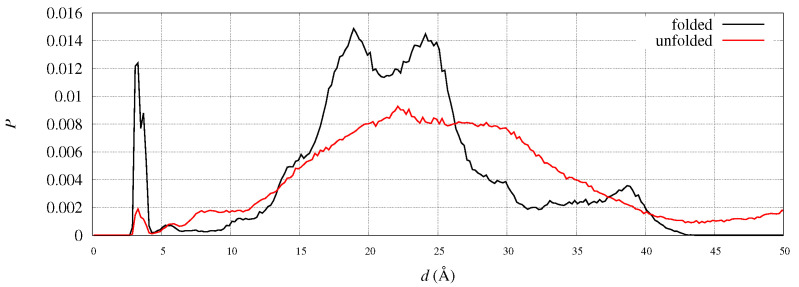
Distribution of the distance between N in the first residue (N(Leu 90)) and C in the last residue (C(Ala 210)) in folded (CV = 14, black curve) and unfolded (CV = 4, red curve) states of the wild-type (WT) sequence.

**Table 1 molecules-27-01955-t001:** Pairs of atoms used in Equation (Equation 16) and related label in parameter *S*. Residues are those of WT sequence. Mutated residues are boldface.

β,1–4		α,1		α,2	
N (Asp 124)	O (Lys 135)	N (Glu 96)	O (Glu 92)	N (Leu 186)	O (Leu 182)
O (Asp 124)	N (Lys 135)	N (Arg 97)	O (Thr 93)	N (Ala 187)	O (His 183)
N (Ser 126)	O (Thr 133)	N (Leu 98)	O (Thr 94)	N (Ala 188)	O (Glu 184)
O (Ser 126)	N (Thr 133)	N (Ala 99)	O (Tyr 95)	N (Glu 189)	O (Leu 185)
N (Val 134)	O (Tyr 143)	N (Glu 100)	O (Glu 96)	N (Leu 190)	O (Leu 186)
O (Val 134)	N (Tyr 143)	N (Glu 101)	O (Arg 97)	N (Thr 191)	O (Ala 187)
N (Leu 132)	O (Ile 145)	N (Thr 102)	O (Leu 98)	N (Lys 192)	O (Ala 188)
O (Leu 132)	N (Ile 145)	N (Leu 103)	O (Ala 99)	N (Ala 193)	O (Glu 189)
N (Val 144)	O (Ser 157)	N (**Asp** 104)	O (Glu 100)		
O (Val 144)	N (Ser 157)	N (Ser 105)	O (Glu 101)		
N (Asn 146)	O (Trp 155)	N (Leu 106)	O (Thr 102)		
O (Asn 146)	N (Trp 155)	N (Ala 107)	O (Leu 103)		
N (Ser 158)	O (Gly 162)	N (Glu 108)	O (**Asp** 104)		
O (Ser 158)	N (Gly 162)	N (**Phe** 109)	O (Ser 105)		
N (Leu 156)	O (Lys 164)	N (Phe 110)	O (Leu 106)		
O (Leu 156)	N (Lys 164)	N (Glu 111)	O (Ala 107)		
N (Ile 154)	O (Tyr 166)	N (Asp 112)	O (Glu 108)		
O (Ile 154)	N (Tyr 166)	N (Leu 113)	O (**Phe** 109)		
		N (Ala 114)	O (Phe 110)		

**Table 2 molecules-27-01955-t002:** Structural parameters *S*, RMSD1, RMSD2, and SASA obtained by imposing average values of CV to 4 and 14, see Section 5 for details. (*) The low $S_{SB}$ is due to the formation of inter-molecular electrostatic interactions responsible of the crystal packing. SASA (Å${}^{2}$) and RMSD (Å) for PDB structures are computed using residues 90-210. RMSD is computed using non-hydrogen backbone atoms.

Variant/State	$S_{\beta ,1–4}$	$S_{\alpha ,1}$	$S_{\alpha ,2}$	$S_{\mathrm{SB}}$	RMSD1	RMSD2	SASA
WT 88-210/PDB 1EKG	15.7	16.1	6.7	1.0 *	−	4.7	6726
WT 42-210/PDB 5KZ5	10.0	11.7	6.6	1.0 *	4.7	−	8909
WT	14	15.9	6.7	5.8	2.5	5.1	7263
D104G	14	15.9	6.6	5.5	2.1	4.9	7145
A107V	14	15.5	6.6	5.7	3.0	5.2	7362
S202F	14	15.9	6.7	5.8	2.3	5.0	7226
S161I	14	15.6	6.6	5.5	2.8	5.3	7313
S181F	14	15.7	6.7	5.6	2.9	5.3	7368
F109L	14	15.7	6.6	5.2	2.4	5.1	7233
Y123S	14	15.7	6.7	5.4	2.2	5.1	7193
W173C	14	15.5	6.7	5.5	2.9	5.1	7368
WT	4	13.8	6.6	6.0	7.6	8.1	8956
D104G	4	13.9	6.7	5.8	7.8	8.1	8910
A107V	4	14.0	6.7	6.2	7.8	8.3	8882
S202F	4	13.6	6.6	5.7	7.6	7.9	9000
S161I	4	14.1	6.7	6.3	7.2	7.8	8837
S181F	4	13.6	6.6	5.9	8.6	8.9	9406
F109L	4	13.7	6.6	6.2	7.4	8.0	8888
Y123S	4	13.8	6.7	6.0	7.6	7.9	8983
W173C	4	14.1	6.6	6.0	7.4	7.9	8781

**Table 3 molecules-27-01955-t003:** Experimental ($\Delta T_{m}$, C${}^{\circ }$) and ΔΔF (kcal/mol) compared to predictions reported in the literature and in this work. Rows are reported in descending order of $\Delta T_{m}$.

	$\Delta T_{m}$	$\Delta \Delta F_{\mathrm{CD}}^{H_{2}O}$	ΔΔF	$\Delta \Delta F_{\mathrm{raw}}$	ΔΔF	ΔΔF
Variant			(I-Mutant)		Equation (Equation 12)	Equation (Equation 10)
	Ref. [15]	Ref. [15]	Ref. [15]	Ref. [18]	This Work	This Work
D104G	3.0	0.21	0.06	−2.50	20.1	−36.9
S202F	−0.3	−0.16	−0.73	49.1	−7.3	−43.4
A107V	−3.0	0.80	0.66	28.1	−114.2	−54.8
S161I	−11.0	−3.35	0.15	−47.7	−53.2	−84.6
S181F	−11.1	−3.11	0.60	−11.6	−38.4	−88.2
F109L	−11.4	−2.09	−0.75	49.8	−21.3	−48.6
Y123S	−14.4	−4.92	−2.46	38.3	−25.2	−49.0
W173C	−	−	−1.40	−68.8	−147.9	−40.4

**Table 4 molecules-27-01955-t004:** Difference in change of energy components (kcal/mol) when imposing average values of CV to 4 and using as reference the imposed average of 14. Equation (Equation 12) is used to replace *H* in Equation (Equation 10) for the free energy decomposition. $U_{int}$ is the sum of stretching, bending and torsional components in Equation (Equation 13); $U_{vdw}$ is the sum of Lennard-Jones interactions for all atomic pairs in the protein; $U_{el}$ is the sum of Coulomb interactions for all atomic pairs in the protein; $U_{solv}=U_{solv,np}+U_{solv,pol}$ (Equation (Equation 12)). See Methods for details.

Variant	$\Delta \Delta U_{\mathrm{int}}$	$\Delta \Delta U_{\mathrm{vdw}}$	$\Delta \Delta U_{\mathrm{el}}$	$\Delta \Delta U_{\mathrm{solv}}$	$\Delta \Delta (U_{\mathrm{el}}+U_{\mathrm{solv}})$
D104G	−0.9	11.0	34.0	−33.6	0.4
S202F	−3.8	10.6	13.4	−16.5	−3.1
A107V	−3.8	−4.6	−47.0	34.0	−13.1
S161I	−3.8	−1.1	−37.0	34.7	−2.3
S181F	−1.6	15.4	33.1	−50.9	−17.8
F109L	−3.9	9.3	−62.1	57.1	−5.0
Y123S	−2.8	11.4	−55.2	46.3	−9.0
W173C	−3.3	−10.0	−72.3	55.9	−16.4

**Table 5 molecules-27-01955-t005:** Block average and error analysis on ΔF (kcal/mol) (calculated at *T* = 400 K and Equation (Equation 12)) and on solvent-accessible surface area (SASA, Å${}^{2}$). Data are for wild-type sequence. Last columns report λ and its error.

*s*	ΔF	Error	SASA	Error	λ	Error
2	729	8	9732	7	1.180391	1.325156
3	653	5	9214	2	0.5549268	2.095248
4	587	4	8956	1	0.2600595	5.996718
5	527	4	8767	1	0.1414134	11.42252
6	468	4	8592	1	0.06856313	16.10580
7	409	4	8421	1	0.01252682	19.44361
8	351	3	8252	1	−0.03635645	21.22864
9	293	3	8085	1	−0.08299696	21.37667
10	235	3	7918	1	−0.1311940	19.86152
11	176	2	7753	1	−0.1856282	16.70221
12	118	2	7587	1	−0.2553848	11.98405
13	59	1	7422	1	−0.3700830	5.981603
14	0	−	7263	2	−0.9076913	0.5297491

**Table 6 molecules-27-01955-t006:** Parameters used in MD simulation for each walker. Temperature *T* is 300 K, except in stage 0 where it is step-wise increased from 0 to 300 K during the first 0.2 ns. We work in the NpT ensemble with the pressure *p* kept constant at 1 bar. Wherever the values of α and *w* are reported, these numbers are used to implement Equation (Equation 3) of Ref. [22]. The bias used at the each stage (when + symbol is present) is obtained as a combination of the biases coming from all the walkers determined in the previous stage. In the last stage (7) the bias is no more updated with deposited gaussian functions. Sampling performed in stage 7 for all walkers provides the metastatistics that will be finally used to compute thermal averages.

Simulation Stage	Ensemble	Bias	α	*w*	*t* (ns)
0	NVT	−	−	−	0.4
1	NPT	−	−	−	8
2	NPT	+	0.00	1	2
3	NPT	+	0.25	1	2
4	NPT	+	0.50	1	2
5	NPT	+	0.75	1	2
6	NPT	+	1.00	0.5	8
7	NPT	+	−	−	10

## Data Availability

All data not reported can be requested to corresponding author.

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
