# Peer review of "Modelling Protein Plasticity: The Example of Frataxin and Its Variants"

_molecules, 2022, doi:10.3390/molecules27061955_

Round 1
Reviewer 1 Report
Botticelli et al. presented a computational work on calculating the effects of protein mutations, by taking frataxin and its pathological mutants as an example, on the change of folding/unfolding free energy using metadynamics with the aid of the maximal constrained entropy method. They showed promising results that the thermal stability differences between the wild-type and mutants could be captured and in reasonable agreement with thermodynamic experiments. I found the enhanced sampling method used in this work is very interesting and could be very useful to other biological problems. Overall, the manuscript is well written and in good quality. But I have a few concerns that I hope they could address before publication:
- By following the idea of altruistic metadynamics which was designed to enhance the sampling of all chemically similar systems at the same time, I thought the authors sampled the WT and eight single-point mutants in one single set of simulations. But by reading the method section, it seems they sampled them separately using multi-walker metadynamics (with 90 walkers for each) instead. Please correct me if I understood it wrong. Otherwise, please make it clearer in the manuscript.
- The discussions on structural deviation and thermal stability (Fig. 2, and 4-9) were all based on meta-statistics (the biased probability) rather than the reweighted statistics corrected by the modulation factors obtained by the maximal constrained entropy method. Please provide a rationale.
- In addition, no error bars for all the probabilities were provided. This made it hard to judge the robustness of the results and conclusions.
- From the meta-statistics profiles, it seems there are multiple intermediates along the folding/unfolding pathways, but with very few discussions.
- The authors specified NpT MD was used here a couple of times. Does the method only work for NpT, but not for other ensembles, such as NVT?
- For better reproducibility of the work, it would be better to open the codes and scripts used in this work, such as in Github.
A few minor points:
1) Line280-281: “A salt-bridge is defined as the formation of a contact between N atoms belonging to positively charged groups and C atoms belonging to carboxylate groups”. Should the C atoms be O atoms?
2) The meta-statistics plots could be organized in a more concise way, such as by combining the subplots in one single plot but with different colors to distinguish the mutants. And it could help if more structural representations of the protein are added.
Author Response
Please, see attached cover letter.

Reviewer 2 Report
Evaluation report in the attached file.

Author Response
Please, see attached cover letter.
